# The Influence of Parathyroidectomy on Osteoporotic Fractures in Kidney Transplant Recipients: Results from a Retrospective Single-Center Trial

**DOI:** 10.3390/jcm11030654

**Published:** 2022-01-27

**Authors:** Ulrich Jehn, Anja Kortenhorn, Katharina Schütte-Nütgen, Gerold Thölking, Florian Westphal, Markus Strauss, Dirk-Oliver Wennmann, Hermann Pavenstädt, Barbara Suwelack, Dennis Görlich, Stefan Reuter

**Affiliations:** 1Department of Medicine D, Division of General Internal Medicine, Nephrology and Rheumatology, University Hospital of Muenster, 48149 Muenster, Germany; anja.kortenhorn@ukmuenster.de (A.K.); katharina.schuette-nuetgen@ukmuenster.de (K.S.-N.); gerold.thoelking@ukmuenster.de (G.T.); florian.westphal@ukmuenster.de (F.W.); dirkoliver.wennmann@ukmuenster.de (D.-O.W.); Hermann.Pavenstaedt@ukmuenster.de (H.P.); Barbara.Suwelack@ukmuenster.de (B.S.); Stefan.Reuter@ukmuenster.de (S.R.); 2Department of Medicine C, Division of Cardiology and Angiology, University Hospital of Muenster, 48149 Muenster, Germany; Markus.Strauss@ukmuenster.de; 3Institute of Biostatistics and Clinical Research, University of Muenster, 48149 Muenster, Germany; Dennis.Goerlich@ukmuenster.de

**Keywords:** kidney transplantation, chronic kidney disease, mineral bone disorder, parathyroidectomy, parathyroid hormone, osteoporosis, bone fractures

## Abstract

Kidney transplant (KTx) recipients are a high-risk population for osteoporotic fractures. We herein aim to identify the role of pre-transplant parathyroidectomy (PTX) and other modifiable factors associated with osteoporotic fractures in KTx recipients. We conducted a retrospective study involving 711 adult patients (4608 patient-years) who were transplanted at our center between January 2007 and June 2015. Clinical data were extracted from patients’ electronic medical records. Different laboratory and clinical parameters for mineral bone disease (MBD) and osteoporosis, including medication, were evaluated. We chose fracture events unrelated to malignancies or adequate trauma as the primary endpoint. Osteoporotic fractures occurred in 47 (6.6%) patients (median 36.7 months, IQR 45.9) after KTx (fracture incidence of 10 per 1000 person-years). Prior to KTx, subtotal PTX was performed in 116 patients (16.3%, median time 4.2 years before KTx, IQR 5.0). Of the patients with fracture (*n* = 47), only one (2.2%) patient had previously undergone PTX. After adjusting for the known fracture risk factors MBD and osteoporosis, PTX remained a protective factor against fractures (HR 0.134, CI 0.018–0.991, *p* = 0.049). We observed a reduced risk for pathological fractures in KTx patients who underwent PTX, independent from elevated parathyroid hormone at the time of KTx or afterwards.

## 1. Introduction

Kidney transplant (KTx) recipients are a high-risk population for osteoporotic fractures, yet their environment combines a number of risk factors that lead to mineral bone disorder (MBD) and osteoporosis. Osteoporotic fractures increase morbidity and mortality in elderly patients; therefore, their prevention is an important issue [1].

Since the majority of patients after KTx show impaired renal function/chronic kidney disease (CKD) and present a long history of end-stage renal disease (ESRD), CKD-related MBD also plays an important role in KTx recipients. It refers to CKD-MBD after kidney transplantation. Both high FGF-23 levels and hyperparathyroidism are present post-transplant, contributing to hypophosphatemia and hypercalcemia [2]. Secondary hyperparathyroidism (sHPT) and uremic toxin accumulation enhance osteoclast-mediated bone resorption with increased release of calcium and phosphate [2].

After KTx, further transplantation-related factors associated with MBD and osteoporosis occur. In addition to electrolyte imbalances (such as hypophosphatemia, hypomagnesemia and hypercalciuria), reduced physical activity, sarcopenia, and immunosuppressive medication, which usually includes corticosteroids and calcineurin-inhibitors (CNI), increase bone resorption [3,4,5]. Therefore, KTx recipients have a significantly lower bone mineral density (BMD) compared to the general population [5].

Treatment of hyperparathyroidism (HPT) and MBD is primarily based on the assessments of laboratory values of phosphate, calcium, PTH level, alkaline phosphatase, and vitamin D. The primary therapeutic approach is drug-based and aims primarily to identify and treat long-term trends. PTX is indicated for ESRD patients with HPT who do not respond adequately to medical or pharmacologic therapy [6].

New pharmacological approaches with calcimimetics have significantly reduced the frequency of PTXs in ESRD patients [7]. However, improvements in mortality rate and bone metabolism with the use of calcimimetics have not been consistently confirmed.

In patients with primary hyperparathyroidism, Yeh et al. demonstrated that PTX is associated with a reduced risk of fractures [8]. In chronic hemodialysis patients with sHPT, Rudser et al. observed a reduced fracture risk in patients after PTX. 

For patients after KTx who have secondary or tertiary HPT, data regarding fracture risk is sparse, and it is unclear to date whether PTX affects the risk of osteoporotic fractures in this particular setting. Thus, we aimed to clarify the effect of PTX and other factors available for retrospective analysis associated with MBD on osteoporotic bone fractures in KTx recipients.

## 2. Materials and Methods

### 2.1. Study Design and Population

We conducted a retrospective study involving 711 adult patients with a follow-up of 4608 patient-years transplanted at our center between January 2007 and June 2015. The follow-up period ended on 25 June 2019. Transplanted patients under the age of 18 years were excluded. Patients were censored for loss of allograft function and death with a functioning allograft. Patient demographic and clinical characteristics were recorded at the time of KTx. Informed written consent was obtained from all patients to collect their data at the time of transplantation. Patient data were anonymized before analysis. Regarding patients with PTX, we only considered those patients who received PTX before KTx. Seven patients in our collective who had PTX surgery after KTx were excluded. This study was performed in accordance with the Declaration of Helsinki and the International Conference on Harmonization Good Clinical Practice guidelines and approved by the local ethics committee (Ethik Kommission der Ärztekammer Westfalen-Lippe und der Medizinischen Fakultät der Westfälischen Wilhelms-Universität, 2014-381-f-N). Data were extracted from the electronic patient records. Induction therapy was chosen according to immunologic risks. One gram mycophenolate mofetil was given twice a day; the dosage was reduced if adverse events occurred. Prednisolone was started with 500 mg intravenously (i.v.) before KTx, followed by 100 mg for three days; then the dose was reduced by 20 mg/day. A dosage of 20 mg/day was maintained until day 30 and then slowly reduced to 5 mg/day. Maintenance immunosuppressive therapy usually consisted of a calcineurin inhibitor (tacrolimus), mycophenolate sodium or mycophenolate mofetil, and prednisolone. m-TOR-inhibitor-based immunosuppression after KTx was chosen only in a minority of patients, usually when they were enrolled in clinical trials.

### 2.2. Analyzed Markers and Parameters

To characterize the bone metabolism of the study subjects, we evaluated different laboratory markers and parameters, including intact PTH (iPTH) (normal range 15.6–65 pg/mL), total serum calcium levels, ionized calcium levels, medication with native and active vitamin D preparations, vitamin D analogues, bisphosphonates, calcimimetics and IgG2-anti-RANKL-antibodies (denosumab).

Each of the laboratory parameters was evaluated three months and one year after KTx. PTH was also evaluated immediately pre KTx. Medication was evaluated one, two, and three years after KTx.

We chose fracture events unrelated to malignancies or adequate trauma as the primary endpoint.

The primary endpoint was ascertained by reviewing all patient files manually. To evaluate a potential association of tacrolimus metabolism and MBD, we calculated the concentration/dose (C/D) ratio three months after KTx. The C/D ratio allows the classification of patients into slow (C/D ratio ≥ 1.05) and fast tacrolimus metabolizers (C/D ratio < 1.05) [9].

### 2.3. Statistical Analysis

The data were analyzed with IBM SPSS Statistics 27 (IBM Corp., Armonk, NY, USA). The results are expressed as a median with interquartile range (IQR) or mean with standard deviation (SD). Non-continuous parameters were analyzed by Fisher’s exact tests and chi-square tests and the continuous parameters were analyzed by a Mann–Whitney U-test and Kruskal–Wallis test, respectively, where appropriate. A *p*-value ≤ 0.05 was considered statistically significant.

Repeated measures ANOVA was applied to compare courses of PTH (before KTx, at 3 months, at 12 months) between different groups.

We analyzed the probability for fracture-free survival after KTx by Kaplan–Meier analysis and a log-rank test. To further, estimate the effect of PTX on the fracture incidence, we considered death as a competing risk. Cumulative incidence functions were estimated for both competing outcomes (death and fracture) and PTX groups were compared using Gray’s test.

Univariate Cox proportional hazard models were fitted to identify the potential predictors for fracture events. Death was considered censored in this analysis; i.e., the model corresponds to a cause-specific hazard model. Variables that were considered statistically noticeable by univariate analysis were used for multivariable Cox proportional hazard regression analysis to identify independent prediction factors for fracture events. Besides these, the two well-known influencing factors on bone mineral disease, female sex and dialysis vintage were added to the multivariable Cox proportional hazard regression analysis.

We analyzed the effect of the estimated glomerular filtration rate (eGFR) at different time points after KTx on fracture risk in Cox proportional hazard models using a landmark approach. We fitted five Cox proportional hazard models, starting at 12 months, 24, 36, 48, and 60 months. Within each analysis, only patients who lived without a fracture at the landmark time point were included with their eGFR values, respectively. Event times were recalculated starting from the landmark time point. Thus, each model evaluates the eGFR as a risk factor for new fractures after the respective time point. Results are presented as a forest plot.

For all Cox proportional hazard models, hazard ratios (HR) and 95% confidence intervals (CI) were reported.

## 3. Results

### 3.1. Baseline Characteristics

Baseline characteristics of the study populations are given in Table 1. Median age at transplantation was 53.0 years (range 17.8–78.4), 430 (60.5%) were male, and 204 (28.7%) received a living donor transplant.

Induction therapy was performed using basiliximab in 588 (83.6%) cases. A total of 37 (5.3%) of the patients received thymoglobuline (Table 1).

### 3.2. Outcome Data

Patients’ eGFR values and landmark univariable Cox proportional hazard regressions for fracture risk depending on eGFR are depicted in Figure 1.

Repeated measures ANOVA to compare the PTH courses is given in Figure 2.

Further outcome data are presented in Table 2.

### 3.3. Fracture Events after KTx

Forty-seven patients (6.6%) from our study cohort suffered from at least one fracture event that was not related to malignancy or adequate trauma. The median time of occurrence of fracture events was 36.7 months (IQR 13.8–59.7 months) after KTx. Interestingly, a fracture occurred in only one PTX patient (Figure 3A). In contrast, the vast majority of the 46 (97.8%) patients with fracture events did not undergo PTX. Mortality was not different between patients with and without PTX (Figure 3B,C).

Levels of PTH were comparable between both groups at day 0 (*p* = 0.158), at 3 months (*p* = 0.848), and at 1 year (*p* = 0.874) (Figure 2).

Patients with fracture events during the follow-up period were significantly older than those without fractures (59.47 ± 11.32 vs. 51.47 ± 14.06, *p* < 0.001). There was no statistical difference between the sexes.

To evaluate the influence of renal function displayed by eGFR on pathological fractures, we compared patients without fractures and those patients who developed fractures for their eGFR courses. Renal function in a course of five years was not consistently associated with fracture incidence; however, it was significantly associated with elevated fracture risk after 24 (*p* = 0.01, HR = 0.973) and 36 months (*p* = 0.034, HR = 0.964), but not after 12 (*p* = 0.086, HR = 0.984), 48 (*p* = 0.246, HR = 0.977) and 60 months (*p* = 0.144, HR = 0.972) (Figure 1). The mean eGFR tended to be lower in the fracture group, especially after years two and three (Figure 1).

PTX courses were not significantly different between these groups pre KTx (*p* = 0.158), after three (*p* = 0.848), and after twelve months (*p* = 0.874). However, PTH tended to be higher in the group without fractures at the timepoint prior to KTx (Figure 2).

A total of 36.6% of patients showed hypophosphatemia (<2.5 mg/dL) three months after KTx without a significant difference between patients with and without fractures (33.3% vs. 36.8%, *p* = 0.502). 

There was no significant association between fracture incidence and the common post-transplant complications BKV-viremia (23.4% vs. 23.3%, *p* = 1.000), CMV-viremia (28.3% vs. 33.5%, *p* = 0.522), NODAT (17.0% vs. 16.2%, *p* = 0.839) and rejection episodes (51.1% vs. 38.6%, *p* = 0.122).

### 3.4. Parathyroidectomy

A total of 112 (15.8%) of the patients underwent subtotal PTX before KTx, whereas 599 patients (84.2%) did not receive PTX. None of the patients underwent PTX surgery after KTx.

PTX was performed at a median time of 4.2 years before KTx (IQR 5.0). Autotransplantation of partial parathyroid tissue was performed in 80 of the 112 patients (71.4%) undergoing PTX (Table 3). There are no significant differences in eGFR between patients with and without PTX at any timepoint.

### 3.5. Medication with Calcimimetics

In total, 47 (6.6%) patients were treated with calcimimetics within the first three years after KTx. Two (4.3%) of these patients had previously undergone PTX. The calcimimetic use in patients with or without fractures (7.0% vs. 7.2%, *p* = 1.000) was comparable.

### 3.6. Levels of Parathyroid Hormone

Median levels of iPTH prior to KTx were 217.0 (329.5) pg/mL, 111.0 (130.5) pg/mL after three months, and 99.4 (104.0) pg/mL after one year. They were not significantly different between patients with and without fractures at any of the three times (*p* = 0.208 prior to KTx, *p* = 0.366 after 3 months, and *p* = 0.906 after one year).

Patients who underwent PTX showed lower PTH levels prior to KTx (37.7 (169.5) vs. 232.0 (353.5) pg/mL, *p* < 0.001), after 3 months (56.0 (111.1)) vs. 117.0 (132.8) pg/mL, *p* < 0.001, Figure 4), and also after one year (57.6 (118.1) vs. 104.0 (92.2) pg/mL, *p* = 0.001).

Before KTx, 75.1% of recipients had iPTH levels within the range recommended by the KDIGO guidelines (2–9-fold above the normal range (15.6–65 pg/mL)), 14.1% of patients showed levels above the range, and 10.7% had levels below the range. For patients after KTx, the ideal PTH levels are not known. At one year after KTx, 23.8% of the patients showed iPTH within the assay range, 71.7% had values above, and 4.5% below [6].

Following a study by Perrin et al. who demonstrated a significant fracture difference between low and high intact PTH (cut-off 130 pg/mL) three months after KTx [10], we tested this cut-off after three months for our collective. However, we could not confirm a significant difference for fracture events between patients with PTH < 130 pg/mL and ≥ 130 pg/mL (see also Appendix A). One patient who showed a fracture event within the first three months (1.4 months after KTx) was excluded from this analysis (Figure 5).

### 3.7. Calcium Levels

For the analysis, whole serum calcium levels as well as levels of ionized calcium were evaluated. Whole serum calcium levels were reduced in PTX patients three months after KTx, *p* < 0.001). The same was true for levels of ionized calcium pre KTx (*p* = 0.0109) and three months after KTx (*p* < 0.001), Table 3.

Between patients with and without fractures, there were neither significant differences in calcium levels at day 0 (*p* = 0.685) nor three months after KTx (*p* = 0.323), Table 2.

### 3.8. Vitamin D Preparations and Bisphosphonates

Vitamin D receptor activators (VDRa) (calcitriol, paricalcitol)) were used more frequently in patients after PTX within the first three years after KTx (55.4% vs. 32.9%, *p* < 0.001). In contrast, we found no differences regarding the use of native vitamin D preparations (cholecalciferol, vitamin D3) between patients with or without PTX (51.7% vs. 49.6%, *p* = 0.749).

We addressed the question of whether the difference in the use of VDRa between patients with and without PTX could be the underlying reason for the reduced fracture rate of PTX patients (Table 3).

However, we could not find any relevant difference in the VDRa treatment of patients with and without fracture (34.9% vs. 39.8%, *p* = 0.629, Table 2).

During follow-up, seven (18.4%) patients with fractures received bisphosphonates. Of these, three patients were prescribed bisphosphonates as secondary fracture prophylaxis after a fracture event. In the remaining four patients, bisphosphonates were used due to diagnosed osteoporosis without a fracture event.

### 3.9. Immunosuppressive Medication

The vast majority of patients studied received a tacrolimus-based immunosuppressive regimen (96.6%). Mean trough levels were 7.86 ± 2.64 ng/mL. The median concentration/dose ratio was 1.3 (IQR 1.08). We did not find an association of steroid use (*p* = 1.000) or tacrolimus use (*p* = 0.670) or trough levels (*p* = 0.881), nor of tacrolimus metabolism (C/D ratio; *p* = 0.187), with fracture events after KTx. Moreover, the tacrolimus trough levels or C/D ratio and PTH levels were not associated.

### 3.10. Multivariate Cox Regression Analysis

To determine whether PTX is independently associated with a lower fracture risk after KTx, adjustment was made for the known risk factors age, sex and dialysis vintage and furthermore for the underlying renal disease, which was tested noticeable in the univariable analysis.

In this analysis, PTX remained a protective factor for fractures (HR 0.134 CI 0.018–0.991, *p* = 0.049). Age was confirmed as the most relevant risk factor (HR 1.051 CI 1.023–1.079, *p* < 0.001) (Table 4).

## 4. Discussion

Bone fractures are a complication that greatly affects quality of life. In addition, immobility caused by fractures increases mortality [11]. KTx recipients have an increased risk of fractures. Recently, Evenepoel et al. observed a fracture incidence of 14.1 per 1000 person-years in 518 KTx recipients within a 5.2-year follow-up [12]. The authors found that low BMD was associated with fractures, independent of classic determinants, including history of fractures. In our larger study with a comparable length of follow-up, the incidence was 10 per 1000 person-years, but unlike the incidental fractures counted by Evenepoel et al., we excluded fractures related to adequate trauma or malignancy.

A recent Cochrane analysis addressed the question of the efficacy of different treatments for the prevention of MBD after KTx. The primary efficacy endpoint considered was bone fracture. It remained uncertain whether, apart from bisphosphonates, any other class of drugs reduced the fracture incidence. In the absence of trial data, the effect of PTX on fracture risk also could not be adequately assessed [13].

In chronic hemodialysis patients, Rudser et al. showed that patients who underwent PTX have a reduced fracture risk. Several mechanisms are discussed as potential mechanisms. First, PTH excess is avoided and therefore high bone turnover lesions are mitigated. Second, PTX-induced hungry bone syndrome, which increases bone mineral uptake, may inherit long-term protective effects on fractures [14]. A protective role of PTX against fracture events has already been shown for primary HPT [8], although the mechanisms in both settings are not comparable.

In 1994, Grotz et al. reported in 100 KTx patients that pretransplant PTX was associated with increased risk of post-transplant fractures [15]. However, this has not been confirmed in subsequent studies, and drugs, surgical techniques as well as indications for PTX have changed since then [16]. Nevertheless, adynamic bone disease, which could result from PTX beside malnutrition, uremic toxins, or CKD-related repressed WNT/β-catenin signaling [17], might have been responsible for the increased fracture risk of patients with PTX in the mentioned study.

In contrast, in the present era of calcimimetics, which probably cannot reduce the fracture risk in ESRD or KTx patients [13,18,19], we herein generate evidence that PTX may act as an independent protective factor against pathologic fractures in KTx recipients with HPT. This association persists when adjusted for the well-known risk factors of age, sex and dialysis vintage, as well as for underlying renal disease, which was associated with fracture events in univariate analysis. Since eGFR at different time points was inconsistently identified as significant risk factor for fractures in univariate Cox proportional hazard regression analysis (after 2 and 3 years, but not after 1, 4, and 5 years, (Figure 1) we did not include this parameter in further analysis. Moreover, the addition of eGFR as reflection of renal function besides age to multivariable analysis seems problematic, because age is already considered in the eGFR estimation formula (CKD-EPI formula) (Levey et al., 2009). Further, age is known to be strongly associated with graft function and outcome in KTx recipients (Legendre et al., 2014). Reasons for this include the fact that living donations, which have a better graft outcome, are more common in younger recipients (Hart et al., 2019), and organs provided under the European senior program (ESP) have lower organ quality because they meet the expanded donor criteria (ECD) (Pascual et al., 2008). 

Isaksson et al. matched a cohort of 590 PTX patients on dialysis or with functioning allograft for assessment of fracture risk [20]. It was observed that PTX reduced the hip fracture risk, but only in female patients compared with non-PTX patients. For this study, the authors distinguished only between KTx and non-KTx patients at the time of PTX. Whether patients received KTx after PTX was not considered. Nevertheless, these results point in the same direction as ours do.

Another study analyzed hypercalcemia control with subtotal PTX compared with cinacalcet use in KTx recipients. Cruzado et al. showed a superior effect of PTX for calcium and PTH normalization and an increase in BMD; but, due to the relatively short follow-up period of twelve months and small patient numbers, they were not able to sufficiently comment on fracture events [21].

Based on data suggesting that hypoparathyroidism after PTX in KTx recipients correlates with a significant decrease of renal function, PTX should be indicated with caution, and the initial treatment approach should be pharmacological with administration of calcimimetics [6,22,23]. This approach was largely followed in our center. Interestingly, Mathur et al. did not observe an association between treatment of sHPT and posttransplant delayed graft function, graft failure, or death, but the proportion of PTX-treated patients in their cohort of 5094 KTx recipients was small (4.5%) [24]. However, our data do not show a significant difference in eGFR courses between patients with and without PTX (Table 3).

In the group of patients in whom a drug approach was followed to lower PTH, it seems trivial that the PTH value remains elevated (Figure 3) [25]. This is paralleled by higher calcium levels and lower phosphate levels (Table 3). Nevertheless, there is also evidence to support a preference for PTX over the use of cinacalcet. In patients with tertiary hyperparathyroidism after KTx, cinacalcet can normalize the serum calcium levels, but unlike PTX, it cannot normalize the PTH levels [26,27]. However, there is no consensus on which PTH value clearly defines post-transplant HPT [28], but PTH levels were found to be an important negative independent predictor of MBD, intriguingly more deleterious than the cumulative dose of corticosteroids or inflammation [5]. In our cohort, one year after KTx the PTH levels decreased 2.2-fold compared to pre KTx. However, one year after KTx, only 23.8% of the patients showed PTH levels within the by KIDGO recommended range for patients with CKD, 71.7% had values above, and 4.5% below (see Results Section 3.6) [6]. This phenomenon has been observed in many studies [28]. Looking at both groups (PTX vs. non-PTX), there was no difference in PTH levels, pointing to the fact that many patients had undergone subtotal PTX and controlled HPT in the other group. Considering that the group without fractures included 17.8% PTX patients (vs. 2.2% in the fracture group), it becomes clear that the average PTH level in the group without fractures was higher in the non-PTX patients and therefore is probably not the decisive influencing factor (Table 3). Nevertheless, there is evidence that persistent HPT could be a risk factor for fractures after KTx. Therefore, it could be important that parathyroid recovery from CKD-induced HPT was incomplete even 1 year after KTx in a large subset of patients (Figure 1). While some studies link PTH-related stimulation of bone turnover to fractures [10], other sources describe (circulating) PTH levels as poorly predictive of underlying bone turnover [29]. In contrast to Perrin et al., who described a PTH above a cut-off of 130 pg/mL measured three months after KTx as a significant risk factor for fracture events in a cohort of 143 KTx recipients with a total of 22 fracture events [10], we could not confirm this observation in our data (Figure 4). Moreover, according to our data, we could not support a clear cut-off for PTH being associated with an increased fracture risk (Appendix A). In agreement with Perrin et al., we did not observe elevated alkaline phosphatase as a marker for bone turnover in the patients with fracture events. However, in contrast to Perrin et.al, we found alkaline phosphatase elevated in those patients without PTX. Nevertheless, it seems possible that bone abnormalities induced by HPT can be explained by the elimination of skeletal resistance to PTH occurring during CKD after renal transplantation.

VDRa (1.25(OH)2D3 (Calcitriol)) are more frequently applied in patients after PTX than in patients without PTX (Table 3). VDRa provide a well-studied protective effect on bone metabolism, whereas the efficacy of nutritional vitamin D preparations, frequently used in non-PTX KTx patients, has not been clearly established yet [30]. However, VDRa use did not differ in patients with and without fractures (Table 2).

In KTx recipients, immunosuppressive therapy is a specific contributing factor to MBD and osteoporosis. In addition to the osteoporotic effects of long-term steroid use [31], CNI medication may also affect bone metabolism. Luo et al. described increased bone resorption in patients with tacrolimus trough levels above 6 ng/mL [32]. Increased bone loss from CNI use has also been demonstrated in a rat model [33]. In contrast, we did not find an association between higher tacrolimus levels or tacrolimus metabolism, represented by the Tac C/D ratio [34], and pathologic fracture events in our cohort. Nevertheless, tacrolimus may influence bone resorption to an extent that is clinically not apparent and is overshadowed by more relevant factors. Recently, the effect of steroids on BMD was prospectively analyzed in de novo KTx patients using steroid minimization protocols. The authors showed that a cumulative methylprednisolone dose of 2.5 ± 0.8 g (after 1 year) and 5.8 ± 3.3 g (after 5 years) caused only limited BMD changes and was predominantly related to remodeling activity rather than corticosteroid exposure [35]. Since treatment per protocol in our center results in exposure of approximately 3.4 g prednisone after 1 year and 10.7 g after 5 years (in an average patient on steroids without rejection), we cannot exclude a relevant influence of steroids on BMD in our cohort. However, steroid use and rejection rates did not differ between the groups.

Interestingly, autosomal-dominant polycystic kidney disease (ADPKD) as an underlying disease for ESRD was associated with the incidence of fractures after KTx in univariate analysis. This fits with the observations of Gitomer et al. who describe a bone metabolism defect in ADPKD patients with CKD stages 1 and 2, although ADPKD was not associated with an increased risk of fractures in ESRD patients [36]. Nevertheless, we could not confirm this association in the multivariate analysis.

This study has several noteworthy limitations. Due to its retrospective nature, it is only hypothesis generating. Since we did not assess additional corticosteroid therapies, which were temporarily applied for the treatment of rejection episodes, nor the VDRa or vitamin D doses or levels, we cannot comment on that and cannot exclude the related effects. Moreover, additional information on FGF23 and BMD would have provided further valuable information on the bone homeostasis and morphology of patients after PTX compared to those without PTX. Nevertheless, these parameters were not routinely assessed in our patients. To at least partially address this limitation, we included known risk factors for lower BMD and fracture risk after KTx into our multivariable model, specifically, age, female sex, dialysis vintage, and cause of ESRD [37], to indirectly account for BMD.

## 5. Conclusions

In conclusion, our study points towards an association between PTX and fracture events after KTx, which is independent from the elevated PTH levels at the time of KTx or afterwards. Therefore, keeping in mind that PTX surgeries were performed considerably prior to KTx, the fracture events might be the result of long-lasting sHPT in the foregoing history of patients without PTX, to a significant extent. As pre-transplant PTX does not influence eGFR after KTx and offers protective effects on fracture risk in our study, our data supports a more generous indication for PTX in ESRD patients with HPT waiting for KTx.

## Figures and Tables

**Figure 1 jcm-11-00654-f001:**
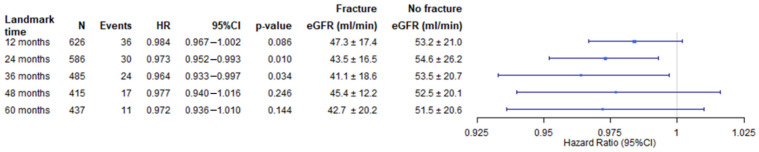
Landmark analysis of univariable Cox proportional hazard regressions for fracture risk depending on eGFR after 12, 24, 36, 48 and 60 months depicted as Forest plot. Mean eGFR values and standard deviations at each time point are given for patients with and without fractures.

**Figure 2 jcm-11-00654-f002:**
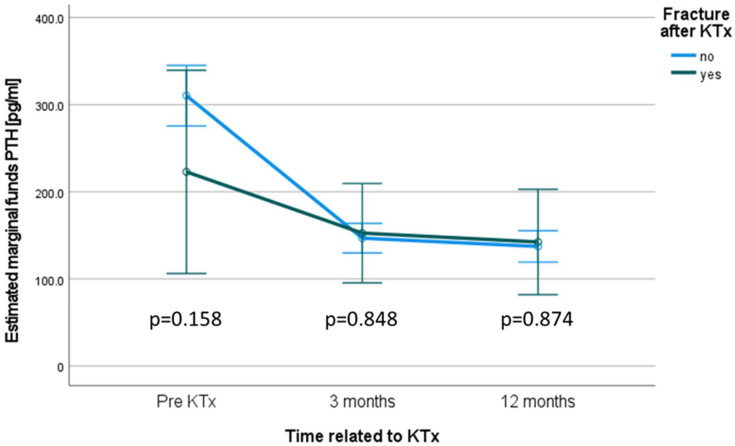
PTH courses in patients with and without fractures. Differences were tested for significance with repeated measures ANOVA. *p*-values are given within the figure.

**Figure 3 jcm-11-00654-f003:**
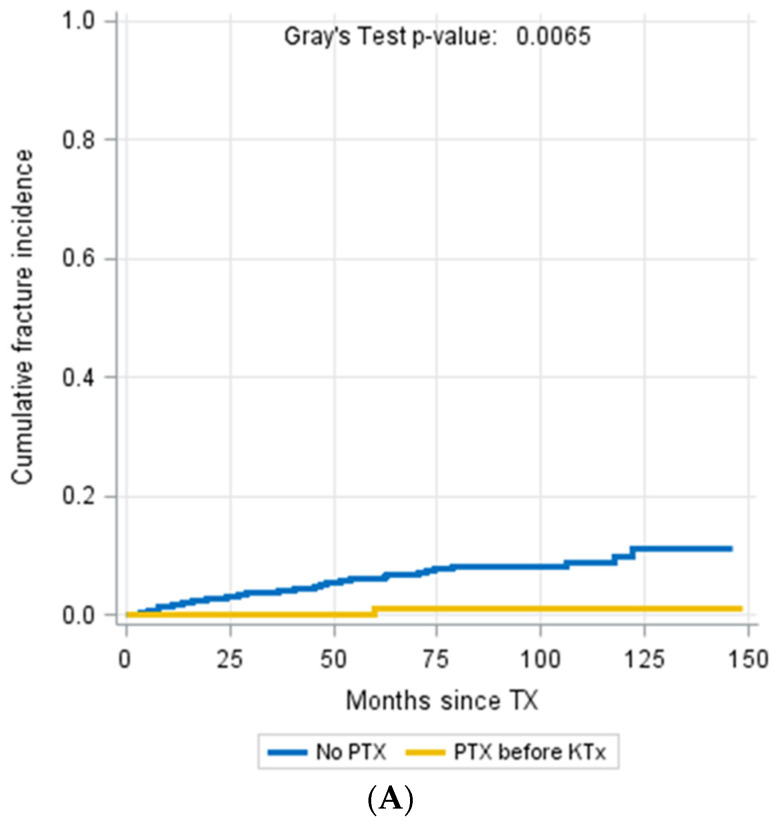
Patients who underwent PTX showed a significantly reduced fracture incidence during the follow-up period (Gray´s test *p* = 0.0065) (**A**). Mortality incidence (Gray´s Test *p* = 0.2574, (**B**)) and fracture-free survival probability were not different between patients with and without fractures (Log-Rank *p* = 0.4703, (**C**)).

**Figure 4 jcm-11-00654-f004:**
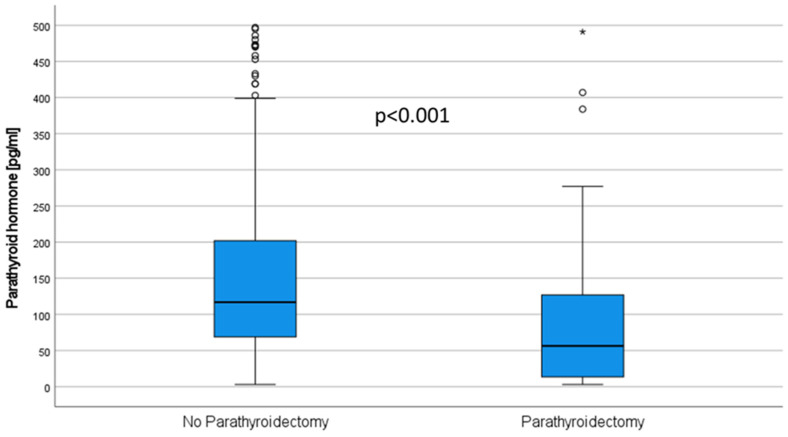
Levels of parathyroid hormone are significantly lower in patients after subtotal PTX compared to non-PTX patients three months after KTx.

**Figure 5 jcm-11-00654-f005:**
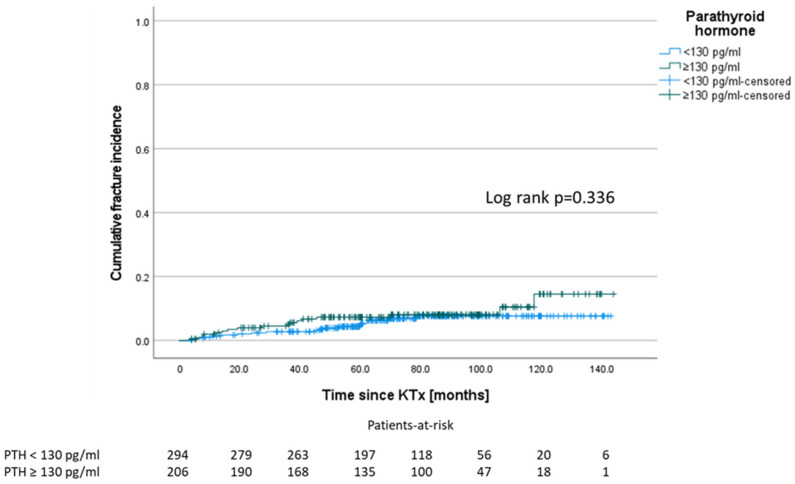
PTH levels ≥ 130 pg/mL after three months are not significantly associated with a higher fracture incidence. One patient, who showed a fracture event within the first three months, was excluded from this analysis.

**Table 1 jcm-11-00654-t001:** Patients’ demographic and clinical characteristics at transplantation.

Variable	All ** (*n* = 711)	Fracture (*n* = 47)	No Fracture (*n* = 664)	*p*-Value
Age at Tx * (years), median (IQR)	53.0 (21.7)	60.8 (11.7)	52.39 (21.9)	0.000 ^a^
Sex male, *n* (%)	430 (60.5%)	22 (46.8%)	406 (61.5%)	0.063 ^b^
mismatch-HLA-A, *n* (%)				1.000 ^b^
none	250 (35.3%)	16 (34.0%)	232 (35.3%)	
1	338 (47.7%)	23 (48.9%)	313 (47.6%)	
2	120 (16.9%)	8 (17.0%)	112 (17.0%)	
mismatch-HLA-B, *n* (%)				0.152 ^b^
none	163 (23.0%)	12 (25.5%)	151 (23.0%)	
1	339 (47.9%)	27 (57.4%)	309 (47.0%)	
2	206 (29.0%)	8 (17.0%)	197 (30.0%)	
mismatch-HLA-DR, *n* (%)				0.607 ^b^
none	178 (25.1%)	12 (25.5%)	164 (25.0%)	
1	340 (48.0%)	20 (42.6%)	320 (48.7%)	
2	190 (26.8%)	15 (31.9%)	173 (26.3%)	
PRA > 85%, *n* (%)	12 (1.7%)	1 (2.1%)	11 (1.7%)	0.566 ^b^
PRA > 5%, *n* (%)	84 (11.8%)	4 (8.5%)	80 (12.2%)	0.641 ^b^
Living donor Tx *, *n* (%)	204 (28.7%)	10 (21.3%)	194 (29.4%)	0.317 ^b^
ABO incompatible Tx *, *n* (%)	40 (5.6%)	0 (0.0%)	40 (6.1%)	0.101 ^b^
Cold ischemia time (hours), median (IQR)	7.93 (9.18)	7.67 (7.03)	7.96 (9.21)	0.779 ^a^
Warm ischemia time (minutes), mean ± SD	32.94 ± 8.33	34.20 ± 8.93	32.82 ± 8.29	0.130 ^a^
Dialysis prior to Tx *, *n* (%)	662 (93.1%)	45 (95.7%)	612 (92.7%)	0.764 ^b^
Dialysis vintage (months) median (IQR)	54.59 (65.23)	54.59 (51.44)	53.82 (66.35)	0.622 ^a^
Previous Tx *, *n* (%)	93 (13.1%)	6 (12.8%)	87 (13.2%)	1.000 ^b^
CMV mismatch D/R, *n* (%)				0.489 ^b^
D^−^/R^−^	122 (17.2%)	6 (13.0%)	116 (17.6%)	
D^−^/R^+^	123 (17.3%)	5 (10.9%)	118 (17.9%)	
D^+^/R^−^	165 (23.2%)	12 (26.1%)	152 (23.0%)	
D^+^/R^+^	300 (42.2%)	23 (50.0%)	274 (41.5%)	
Initial immunosuppression, *n* (%)				
Initial steroid use	700 (98.5%)	47 (100%)	649 (98.3%)	1.000 ^b^
Initial MMF * use	683 (96.1%)	44 (93.6%)	635 (96.2%)	0.423 ^b^
Initial CyA * use	24 (3.4%)	2 (4.3%)	22 (3.3%)	0.670 ^b^
Initial tacrolimus use	687 (96.6%)	45 (95.7%)	638 (96.7%)	0.670 ^b^
Initial mTOR * inhibitor use	29 (4.1%)	3 (6.4%)	26 (3.9%)	0.433 ^b^
Diagnosis of ESRD, *n* (%)				0.004 ^c^
Hypertension	55 (7.7%)	1 (2.1%)	54 (8.2%)	
Diabetes	43 (6.0%)	3 (6.4%)	40 (6.1%)	
Polycystic kidney disease	104 (14.6%)	15 (31.9%)	88 (13.3%)	
Obstructive Nephropathy	35 (4.9%)	2 (4.3%)	33 (5.0%)	
Glomerulonephritis	228 (32.1%)	6 (12.8%)	221 (33.5%)	
FSGS *	32 (4.5%)	3 (6.4%)	29 (4.4%)	
Interstitial nephritis	36 (5.1%)	2 (4.3%)	33 (5.0%)	
Vasculitis	23 (3.2%)	4 (8.5%)	19 (2.9%)	
Other	102 (14.3%)	6 (12.8%)	96 (14.5%)	
Unknown	53 (7.5%)	5 (10.6%)	47 (7.1%)	

^a^ Mann–Whitney U-test, ^b^ Fisher´s exact test, ^c^ Chi square test, * Abbreviations: Tx: transplantation; HLA: human leukocyte antigen; PRA: panel reactive antibodies; MMF: mycophenolate, mofetil; CyA cyclosporine A; FSGS: focal segmental glomerulosclerosis, ** missing values: if total numbers are below 711, single values were not available in electronic patient records.

**Table 2 jcm-11-00654-t002:** Characteristics of patients stratified by fracture events.

	All (*n* = 711)	Fracture (*n* = 47)	No Fracture (*n* = 664)	*p*-Value
Alkaline phosphatase at M3 (u/L) median (IQR)	83.50 (43)	84.50 (41)	83.00 (44)	0.696 ^a^
Whole serum calcium at prior to KTX (mmol/L) median (IQR)	2.29 (0.29)	2.33 (0.3)	2.29 (0.29)	0.539 ^a^
Ionized calcium at prior to KTX (mmol/L) median (IQR)	1.18 (0.12)	1.19 (0.09)	1.18 (0.12)	0.264 ^a^
Whole serum calcium at M3 (mmol/L) median (IQR)	2.39 (0.24)	2.38 (0.29)	2.40 (0.24)	0.685 ^a^
Ionized calcium at M3 (mmol/L) median (IQR)	1.27 (0.14)	1.25 (0.14)	1.27 (0.13)	0.323 ^a^
Serum phosphate at M3 (mg/dL) median (IQR)	2.70 (1.1)	2.70 (1.2)	2.70 (1.1)	0.502 ^a^
Calcium phosphate product at M3 ((mg/dL)/(mmol/L))	6.55 (2.23)	6.48 (2.35)	6.55 (2.22)	0.579 ^a^
Parathyroid hormone prior to KTX (pg/mL) median (IQR)	217.0 (329.5)	188.0 (209.8)	222.0 (333.6)	0.244 ^a^
Parathyroid hormone at M 3 (pg/mL) median (IQR)	111.0 (130.5)	121.5 (174.1)	108.0 (126.6)	0.322 ^a^
Parathyroid hormone at M 12 (pg/mL) median (IQR)	99.4 (104.0)	95.2 (114.4)	99.7 (102.6)	0.962 ^a^
Use of calcimimetics within Y1-3, *n* (%)	47 (7.2%)	3 (7.0%)	44 (7.2%)	1.000 ^b^
Use of bisphosphonates within Y1-3, *n* (%)	22 (3.3%)	7 (16.3%)	15 (2.5%)	<0.001 ^b^
Use of nutritional Vitamin D within Y1-3, *n* (%)	355 (54.0%)	26 (60.5%)	327 (53.5%)	0.430 ^b^
Use of VDR activators within Y1-3, *n* (%)	259 (39.4%)	15 (34.9%)	243 (39.8%)	0.629 ^b^
Parathyroidectomy prior to KTX, *n* (%)	112 (15.8%)	1 (2.1%)	111 (16.7%)	0.003 ^b^
Death censored allograft survival, *n* (%)	630 (88.6%)	37 (78.7%)	589 (89.2%)	0.053 ^b^
Overall Graft Survival, *n* (%)	568 (79.9%)	30 (63.8)	535 (81.1)	0.008 ^b^
NODAT, *n* (%)	116 (16.3%)	8 (17.0%)	107 (16.2%)	0.839 ^b^
BK viremia, *n* (%)	165 (23.2%)	11 (23.4%)	154 (23.3%)	1.000 ^b^
CMV viremia, *n* (%)	236 (33.2%)	13 (28.3%)	221 (33.5%)	0.520 ^b^
Rejection yes, *n* (%)	280 (39.4%)	24 (51.1%)	255 (38.6%)	0.122 ^b^
Tacrolimus use at M12, *n* (%)	331 (76.8%)	12 (42.9%)	317 (79.1%)	<0.001 ^b^
Tacrolimus C/D ratio at M3, median (IQR)	1.30 (1.08)	1.43 (1.48)	1.30 (1.09)	0.187 ^a^
Tacrolimus trough levels (ng/mL) at M3, mean ± SD	7.86 ± 2.64	7.91 ± 2.65	7.85 ± 2.63	0.881 ^a^
Steroid use at M12	406 (57.1%)	27 (96.4%)	378 (94.0%)	1.000 ^b^
Fracture Localisation				
Extremity		25 (53.2%)		
Femoral neck		6 (12.8%)		
Pelvis		5 (10.6%)		
Vertebrae		8 (17.0%)		
Thorax (rips, sternum, clavicle)		3 (6.4%)		

^a^ Mann–Whitney U-test, ^b^ Fisher’s exact test, Abbreviations: VDR: vitamin D receptor; CI: confidence interval; NODAT: New-onset diabetes after transplantation; crea: creatinine; CMV: cytomegalovirus; BK: BK-Polyomavirus; Tac: Tacrolimus; C/D ratio: concentration/dose ratio.

**Table 3 jcm-11-00654-t003:** Characteristics of patients stratified by PTX.

	Patients with Parathyroidectomy before KTX (*n* = 112)	Patients without Parathyroidectomy before KTX (*n* = 599)	*p*-Value
PTX with partial autotransplantation, *n* (%)	80 (71.4 %)	-	-
Parathyroid hormone at KTx (pg/mL), median (IQR)	37.7 (169.5)	232.0 (353.5)	<0.001 ^a^
Parathyroid hormone at M3 (pg/mL) median (IQR)	56.0 (111.1)	117.0 (132.8)	<0.001 ^a^
Parathyroid hormone at Y1 (pg/mL) median (IQR)	57.6 (118.1)	104.0 (92.2)	<0.001 ^a^
Alkaline phosphatase at M3 (u/L) median (IQR)	77.0 (41)	85.0 (45)	0.035 ^a^
eGFR at Y1 (mL/min/1.73 m^2^) (±SD)	53.6 (±20.0)	55.5 (±20.7)	0.763 ^a^
eGFR at Y3 (mL/min/1.73 m^2^) (±SD)	54.6 (±20.6)	55.2 (±20.3)	0.747 ^a^
eGFR at Y5 (mL/min/1.73 m^2^), mean (± SD)	50.9 (±22.0)	51.5 (±20.4)	0.868 ^a^
Whole serum calcium at prior to KTX (mmol/L) median (IQR)	2.26 (0.38)	2.29 (0.28)	0.379 ^a^
Ionized calcium at prior to KTX (mmol/L) median (IQR)	1.18 (0.11)	1.18 (0.11)	0.010 ^a^
Whole serum calcium at M3 (mmol/L), median (IQR)	2.26 (0.32)	2.41 (0.22)	<0.001 ^a^
Ionized calcium at M3 (mmol/L), median (IQR)	1.22 (0.26)	1.28 (0.13)	<0.001 ^a^
Serum phosphate at M3 (mg/dL), median (IQR)	2.95 (1.5)	2.70 (1.0)	0.001 ^a^
Calcium phosphate product ((mg/dL)/(mmol/L)) at M3, median (IQR)	6.62 (2.39)	6.54 (2.20)	0.237 ^a^
Use of calcimimetics within Y1-3, *n* (%)	2 (1.7%)	45 (7.5%)	0.011 ^b^
Use of bisphosphonates within Y1-3, *n* (%)	4 (3.6%)	17 (2.8%)	0.76 ^b^
Use of nutritional Vitamin D within Y1-3, *n* (%)	58 (51.7%)	297 (49.6%)	0.749 ^b^
Use of VDR activators within Y1-3, *n* (%)	62 (55.4%)	197 (32.9%)	<0.001 ^b^

^a^ Mann–Whitney U-test, ^b^ Fisher´s exact test, Abbreviations: PTX: parathyroidectomy; eGFR: estimated glomerular filtration rate, calculated using CKD-EPI formula; KTx: kidney transplantation; VDR: vitamin D receptor.

**Table 4 jcm-11-00654-t004:** Multivariable Cox regression for factors associated with fractures after KTx.

Variable	Hazard Ratio	95% CI	*p*-Value
Age at KTx	1.051	1.023–1.079	<0.001
Dialysis vintage	0.999	0.991–1.007	0.851
Female sex	1.692	0.919–3.115	0.091
Underlying renal disease	-	-	0.111
Parathyroidectomy	0.134	0.018–0.991	0.049

Abbreviations: CI: confidence interval; KTx: kidney transplantation.

## Data Availability

The datasets generated during and/or analyzed during the current study are available from the corresponding author on reasonable request.

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
