# Peer review of "The Influence of Parathyroidectomy on Osteoporotic Fractures in Kidney Transplant Recipients: Results from a Retrospective Single-Center Trial"

_jcm, 2022, doi:10.3390/jcm11030654_

Round 1

Reviewer 1 Report

Jehn et al. reported the influence of parathyroidectomy on osteoporotic fractures in kidney transplant recipients. This manuscript is well-written, but I have some concerns.

Major>

  1. As the authors mentioned in the limitation section, information on BMD was missing. As osteoporosis is one of the major risk factors of osteoporotic fracture, the incidence and severity of osteoporosis should be considered in analyses for the fracture risk evaluation. Perrin et al. (reference 10) also indicated that pretransplant osteopenia increased the risk of fracture by HR of 2.792 (95% CI 1.074-7.258).

Minor>

  1. Figure 2 showed that PTH level in the no-fracture group was higher than that in the fracture group (line 259-260). Please check it.
  2. Please describe the full term before the abbreviation and use the same abbreviation for the same words. For example, "sHPT" in line 52 needs a full-term description. "KT" in line 59 may be changed to "KTx".

Author Response

Specific feedback Reviewer 1:

We are grateful to the reviewer for her/his critical comments and questions which helped us to considerably improve the manuscript.

Comment 1:

Major

“As the authors mentioned in the limitation section, information on BMD was missing. As osteoporosis is one of the major risk factors of osteoporotic fracture, the incidence and severity of osteoporosis should be considered in analyses for the fracture risk evaluation. Perrin et al. (reference 10) also indicated that pretransplant osteopenia increased the risk of fracture by HR of 2.792 (95% CI 1.074-7.258).”

Response:

We appreciate this valuable objection. We apologize that we are not able to provide information on the patients´ BMD because it is not routinely evaluated at our center. Nevertheless, we included known risk factors for lower BMD and fracture risk after kidney transplantation, specifically age, female sex and dialysis vintage [1], in our multivariable model to indirectly account for patients´ BMD as the mentioned parameters are strongly correlated to lower BMD and osteoporosis. In addition, we analyzed medications typically prescribed in this context such as native and active vitamin D preparations, vitamin D analogues, bisphosphonates, and IgG2-anti-RANKL-antibodies. Thereby, we attempted to indirectly account for information on patients` BMD as accurately as possible.

We added a comment on that in the limitations section.

  1. Iseri, K.; Carrero, J.J.; Evans, M.; Felländer-Tsai, L.; Berg, H.E.; Runesson, B.; Stenvinkel, P.; Lindholm, B.; Qureshi, A.R. Fractures after kidney transplantation: Incidence, predictors, and association with mortality. Bone 2020, 140, 115554, doi:10.1016/j.bone.2020.115554.

Comment 2:

Minor

“Figure 2 showed that PTH level in the no-fracture group was higher than that in the fracture group (line 259-260). Please check it.”

Response:

We are very grateful for this consideration. We have reviewed it and revised it in our manuscript. The PTH level before KTx was higher in the no-fracture group than in the fracture group, although this was not statistically significant.

Comment 3:

Minor

Please describe the full term before the abbreviation and use the same abbreviation for the same words. For example, "sHPT" in line 52 needs a full-term description. "KT" in line 59 may be changed to "KTx"”.

Response:

Thank you for this consideration. We have revised our manuscript carefully so that all abbreviations are preceded by a full-term description.

Reviewer 2 Report

711 patients who understand kidney transplant over a 8 year period. They found PTx to be protective in post-transplant patients.

Would changed parathyroidectomized to “underwent parathyroidectomy”.

What were calcium levels prior to kidney transplant.? The current analysis compares secondary and tertiary HPT patients, as well as medically-refractory 2ndry HPT (those who has PTx).  This leads the analysis to be subject to bias.  Was adjusted analysis or sub-analysis performed?

Author Response

Specific feedback Reviewer 2:

We are grateful to the reviewer for her/his critical comments and questions which helped us to considerably improve the manuscript.

Comment 1:

“Would changed parathyroidectomized to “underwent parathyroidectomy””.

Response:

We are thankful for this remark. We revised all “parathyroidectomized” to the suggested term “underwent parathyroidectomy”.

Comment 2:

“What were calcium levels prior to kidney transplant.? The current analysis compares secondary and tertiary HPT patients, as well as medically-refractory 2ndry HPT (those who has PTx).  This leads the analysis to be subject to bias.  Was adjusted analysis or sub-analysis performed?”

Response:

Thank you for bringing up this important point. We assessed the suggested calcium data and have included them in the revised manuscript. Pretransplant calcium levels were not statistically different in patients with and without fractures. (whole serum calcium p=0.539, ionized calcium p=0.264). We added these values to table 2 in our manuscript.

Expectedly, ionized calcium levels were significantly lower in patients who underwent parathyhroidectomy compared to those who did not (p=0.010). We added this information to table 3 in our manuscript.

Unfortunately, analysis of calcium levels seems very susceptible to bias. Besides the medical treatment with the calcimimetic cinacalcet, nutrition, parathyroidectomy as well as the treatment with loop or thiazide diuretics and the medication with Vitamin D and/or Vitamin D-receptor activators influence calcium levels. Therefore, we tested for the association of parathyroid hormone levels prior to KTx, at 3 months and 12 months after KTx with the fracture incidence. None of these three parameters was significantly associated with the occurrence of fractures in univariable analyses. Parathyorid hormone (PTH) levels might be the most appropriate measure which considers secondary and tertiary HPT as well as pharmacological treatment and medically-refractory secondary HPT with parathyroidectomy to a reasonable extent. PTH levels were not statistically different at one of the three time points between patients with or without fractures (table 2 in our manuscript).

Round 2

Reviewer 1 Report

This manuscript is well revised. I recommend to accept it for publication. Thanks for your effort.